# FreeRide: Harvesting Bubbles in Pipeline Parallelism

## Abstract

The occurrence of bubbles in pipeline parallelism is an inherent limitation that can account for more than 40% of the large language model (LLM) training time and is one of the main reasons for the underutilization of GPU resources in LLM training. Harvesting these bubbles for GPU side tasks can increase resource utilization and reduce training costs but comes with challenges. First, because bubbles are discontinuous with various shapes, programming side tasks becomes difficult while requiring excessive engineering effort. Second, a side task can compete with pipeline training for GPU resources and incur significant overhead. To address these challenges, we propose FreeRide, a system designed to harvest bubbles in pipeline parallelism for side tasks. FreeRide provides programmers with interfaces to implement side tasks easily, manages bubbles and side tasks during pipeline training, and controls access to GPU resources by side tasks to reduce overhead. We demonstrate that FreeRide achieves about 8% average cost savings with a negligible overhead of about 1% for typical long training times of LLMs while serving model training, graph analytics, and image processing side tasks.

## 1 Introduction

Large language models (LLMs) are usually trained on GPUs. As these models continue to increase in size, their GPU memory requirements can easily outstrip the capacity of a single GPU (Zhang et al., 2022). Consequently, to accommodate this increase in size and to boost training performance, it is a common practice to parallelize LLM training across multiple GPUs distributed over several servers.

Pipeline parallelism is a prevalent training paradigm for LLMs using multiple GPUs. In this paradigm, the model is divided into multiple stages which are distributed across different GPUs. During training, the forward propagation (FP) and backward propagation (BP) of different input data are carried out in parallel by the pipeline training system at each stage. The pipeline training system schedules these operations in each epoch to train LLMs (Liu et al., 2023; Qi et al., 2024).

An inherent limitation of pipeline parallelism is *bubbles* — periods in pipeline training where the GPU stays idle due to unsatisfied dependencies between FP and BP operations (Liu et al., 2023). Experimentally, we observe that bubbles can constitute 42.4% of the pipeline execution time, which results in significant under-utilization of GPU resources used to accelerate pipeline training. Similar levels of under-utilization have also been reported in other studies (Zhang et al., 2022).

GPUs are crucial resources, especially those high-end models required for training LLMs (Zhang et al., 2022). To enhance utilization, prior work has explored reducing bubbles by improving how FP and BP operations are interleaved (Fan et al., 2021; Liu et al., 2023). These approaches are effective for intra-epoch bubbles because they change how operations are interleaved within a pipeline epoch. However, they do not remove the inter-epoch bubbles that occur before and after a pipeline epoch. Prior work has also proposed to decouple the computation of gradients for the input and model weights to mitigate inter-epoch bubbles (Qi et al., 2024). However, they increase the size of activations, exacerbating GPU memory consumption, a common bottleneck in training LLMs.

Given the difficulty and overhead incurred in eliminating these bubbles, an alternative approach is to acknowledge their existence and utilize them by running additional workloads on a GPU. For example, Bamboo and PipeFisher implement procedures that enhance pipeline training and run them during bubbles (Thorpe et al., 2023; Osawa et al., 2023). However, they only target specialized

procedures that are tightly coupled with pipeline training, requiring the training system and the procedures to be highly customized. Consequently, they cannot be used for generic GPU workloads.

In this paper, we present FreeRide, a holistic system to harvest bubbles in pipeline parallelism to serve extra GPU workloads as *side tasks*. There are two main challenges that FreeRide has to overcome. First, customizing side tasks for bubbles of various *shapes*, i.e., their duration and available GPU memory, requires enormous programming effort. Second, LLM training requires high-end GPUs that are expensive and in high demand. If side tasks interfere with the main pipeline training workload, e.g., accessing more GPU resources than bubbles can provide, they will slow down pipeline training and significantly increase training costs.

Our approach to overcoming the programming complexity is based on the observation that many GPU workloads naturally consist of small, repetitive steps, such as the epochs in model training that repeatedly load data and update model weights. To reduce the programming effort, FreeRide introduces a framework that abstracts away the implementation details of side tasks, allowing programmers to adapt various side tasks to fit into the bubbles. The key idea is to represent the life cycle of a side task, from its process creation to termination, as states in a state machine. FreeRide provides two sets of unified interfaces — the iterative interface that features lower performance overhead, and the imperative interface that features better versatility. They facilitate the implementation of side tasks as state transitions with little engineering effort. FreeRide manages side tasks through these interfaces and serves them during bubbles.

FreeRide limits the GPU resource consumption of side tasks through the automated side task profiler and the side task manager. The side task profiler first captures the key performance characteristics of the newly implemented side tasks. The side task manager coordinates a group of side task workers, one for each GPU in the platform, and assigns each of the side tasks to one of the workers based on the characteristics. During pipeline training, bubbles are reported to the side task manager from FreeRide-instrumented pipeline training system. The side task manager starts side tasks when the bubble period begins and pauses them when the bubble ends. A side task worker deploys each side task on top of CUDA MPS (Nvidia, 2024d) to limit its GPU memory consumption and uses Docker (Bernstein, 2014) for isolation. These components work collaboratively to serve side tasks during bubbles, achieving a low performance overhead on the primary pipeline training workload.

In summary, FreeRide is a holistic solution that manages and serves the side task by leveraging bubbles in pipeline training, while maintaining minimal performance overhead and requiring low programming effort. We evaluate FreeRide by deploying it to run side tasks alongside DeepSpeed that runs pipeline training (Rasley et al., 2020). We measure the time increase of pipeline training as the performance overhead caused by side tasks. As the throughput of different side tasks is not directly comparable with the pipeline training workload, we use the cost of GPUs as a unified metric, i.e., the cost of the extra execution time from co-locating side tasks with pipeline training vs. the cost saved from running side tasks that otherwise would run on dedicated lower-tier GPUs.

The contributions of this paper are as follows:

- We study the bubbles in pipeline parallelism, present their various shapes in terms of duration and available GPU memory, and demonstrate their potential for side tasks.

- We present the programming framework and interfaces of FreeRide based on a state machine abstraction to implement generic side tasks with little engineering effort.

- We evaluate FreeRide with model training, graph analytics, and image processing side tasks to demonstrate FreeRide's effectiveness in harvesting bubbles in pipeline parallelism while reducing performance overhead.

- By serving side tasks based on the iterative interface, FreeRide achieves an average *cost savings* of 7.8% with a low performance overhead of 1.1%. This is significantly better than using CUDA MPS (Nvidia, 2024d) directly to co-locate the tasks, which results in a 4.5% *cost increase* and 48.7% overhead. When handling a mix of these 3 types of side tasks, FreeRide achieves 10.1% cost savings with a 1.1% overhead.

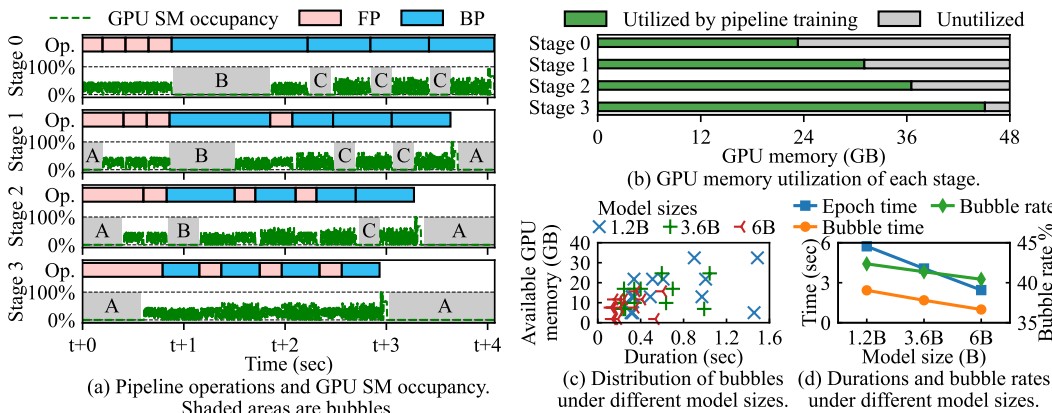

Figure 1: A pipeline training epoch in DeepSpeed and statistics of bubbles for different model sizes.

## 2 BACKGROUND AND MOTIVATION

In this section, we first introduce the underutilization issues due to bubbles in pipeline parallelism and then the motivation for utilizing the bubbles to execute generic workloads.

### 2.1 PIPELINE PARALLELISM AND BUBBLES

Pipeline parallelism is a commonly used scheme to train LLMs that exceed the memory capacity of a single GPU (Rasley et al., 2020; Shoeybi et al., 2020). There are periods in pipeline training when the GPU streaming multiprocessor (SM) occupancy is low, as depicted by the green curves in Figure 1(a). We refer to these periods as *bubbles* in the pipeline, marked as shaded areas. Bubbles inherently exist in pipeline parallelism and occur repetitively throughout training, as they are fundamentally caused by unsatisfied dependencies between FP and BP operations (Liu et al., 2023). In the example of Figure 1, Stage 1 must wait for input from Stage 0 before executing its first FP operation, creating a bubble in Stage 1 that starts from $t + 0$.

To study bubbles in pipeline parallelism, we train a 3.6B-parameter LLM adapted from GPT2-XL (Radford et al., 2019; Choi et al., 2023; Karpathy, 2024) using DeepSpeed (Rasley et al., 2020) on a 4-GPU server (detailed setup in Section 4.1). The training is deployed as a 4-stage pipeline, and each stage takes one GPU as shown in Figure 1. Overall, we observe that bubbles exhibit different characteristics across all 4 stages.

### 2.1.1 BUBBLE CATEGORIZATION

We categorize the bubbles into 3 types based on their positions in the training pipeline and causes.

• **Type-A bubbles** appear at the start and end of each epoch in all stages except for the first stage. They are due to cascading dependencies between operations at the start and end of an epoch. When an epoch starts, the FP operations start at Stage 0, while all other stages wait for input from preceding stages to start their first FP operation. Likewise, at the end of an epoch, the last BP operation starts at Stage 3 and all other stages wait for their succeeding stages to start their last BP operation.

• **Type-B bubbles** occur in the middle of each epoch on all stages except the last one. They are caused by dependencies between interleaved FP and BP operations. Once the first FP operation reaches the last stage, all previous stages must wait for the corresponding BP operation before they can proceed with other operations, which causes Type-B bubbles.

• **Type-C bubbles** also occur in the middle of each epoch. Since BP operations typically take longer than FP operations (Zheng et al., 2022), interleaved yet unaligned FP and BP operations create bubbles in each stage except the last. For instance, in Figure 1(a), when Stage 2 finishes its third BP operation, it must wait for input to its fourth BP operation, which is still being processed in Stage 3, causing a type-C bubble.

**Bubble Duration.** In our training setup, the duration of a bubble ranges from 0.22 to 1.04 seconds, depending on its type and stage. The duration increases for Type-A bubbles but decreases for Type-B bubbles from Stage 0 to Stage 3. This is because of the cascading dependency from Stage 3 to Stage 0 for Type-A bubbles and from Stage 0 to Stage 3 for Type-B bubbles. For example, a Type-B bubble at Stage 2 is due to an unfinished BP operation at Stage 3, with the same bubble at Stage 1 caused by Stage 2. The accumulated time to satisfy dependencies elongates Type-A or Type-B bubbles at later stages. However, Type-C bubbles are caused by unaligned FP and BP operations. Therefore, they have a short duration and do not exhibit the same stage-dependent variations.

**Available GPU Memory.** Determined by the stage, the available GPU memory of a bubble ranges from less than 3 GB to more than 20 GB in our setup. As shown by Figure 1(b), within a stage, the GPU memory consumption of pipeline training remains the same. Thus, the bubbles within the same stage have the same amount of available GPU memory. Because the later stages consume less GPU memory to store activations used by BP operations (Liu et al., 2023), the available GPU memory increases from Stage 0 to Stage 3.

We further study pipeline training of models of different sizes. As shown in Figure 1(c), bubble shapes differ. Overall, bubbles in larger LLMs have less available memory and shorter duration, but the distributions are similar as training follows the same pipeline schedule. Even larger models do not eliminate bubbles as they inherently exist. Under the same configuration, the characteristics of bubbles remain the same during training as epochs are repetitive and stable.

### 2.1.2 BUBBLE RATE

Besides the bubble shape, we evaluate the overall *bubble rate*, i.e., the total bubble time over pipeline training time. When the model size increases from 1.2B to 6B parameters, as shown in Figure 1(d), both the per-epoch time in pipeline training and the total per-stage bubble time decrease. Therefore, the bubble rate drops only slightly from 42.4% to 40.4%. We also evaluate a larger micro-batch number, i.e., an increase from 4 (used in Figures 1) to 8. The bubble rate drops to 26.2% as each epoch takes longer.

Prior work has focused on reducing bubbles in pipeline parallelism. One approach is designing different ways of interleaving FP and BP operations (Fan et al., 2021; Liu et al., 2023). This approach optimizes the scheduling strategies and interleaves FP and BP operations within an epoch. Therefore, they are effective for Type-B and Type-C bubbles that appear inside an epoch but not for Type-A bubbles. Another approach is to reduce Type-A bubbles by decoupling the computation of gradients for the input and the model weights (Qi et al., 2024). This comes at a cost of higher GPU memory usage due to the extra activation storage, exacerbating the GPU memory bottleneck in LLM training. Despite these efforts, none of the approaches fully eliminate bubbles in pipeline training.

### 2.2 UTILIZING BUBBLES

The difficulties in mitigating these bubbles motivate an alternative approach — acknowledging their existence and leveraging their resources for benefits. GPUs used for training are generally compute-rich, with sufficient GPU memory available during the bubbles to accommodate other GPU workloads. Therefore, bubbles can be used to run workloads that otherwise require dedicated GPUs.

Previous work attempts to leverages such GPU resources in two ways, (1) by implementing dedicated procedures, and (2) by transparent GPU sharing. Bamboo and PipeFisher implement procedures that enhance pipeline training and run them during bubbles (Thorpe et al., 2023; Osawa et al., 2023). However, they tightly couple the pipeline training system with the specialized procedures that involves complicated implementation especially since such customization should consider various bubble shapes — with durations ranging from 0.22 to 1.04 seconds, and available GPU memory from less than 3 GB to more than 20 GB on each GPU (Section 2.1). Therefore, they cannot be used for generic GPU workloads.

Transparent GPU sharing does not require complicated implementation to adapt GPU workloads to bubbles (Nvidia, 2024c;d; Strati et al., 2024). However, they are oblivious of the patterns of pipeline parallel training and bubbles, and can cause significant performance overhead on the high-priority and high-cost pipeline training (Section 4). In addition, some GPU sharing work is tailored for certain software toolchain, which significantly limits its versatility (Strati et al., 2024).

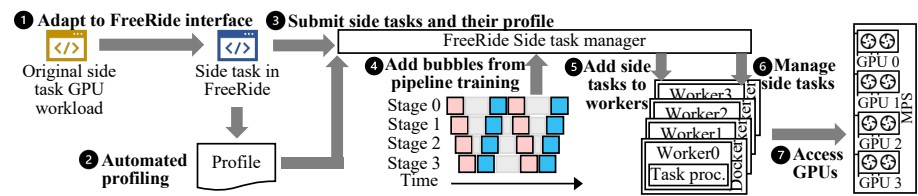

Figure 2: Overview of FreeRide.

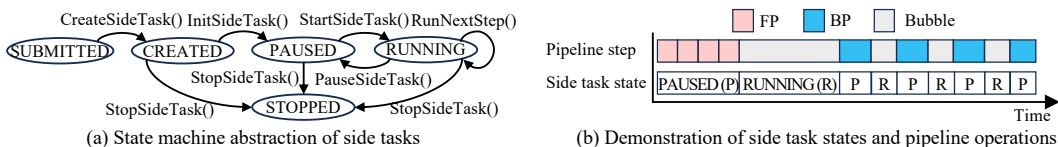

(a) State machine abstraction of side tasks      (b) Demonstration of side task states and pipeline operations

Figure 3: State transitions in a side task program.

In this work, we aim to *make bubble resources available to generic workloads, allowing for a programmable and efficient use of bubbles, while minimizing the overhead of side tasks on the high-priority pipeline training.* We identify two major challenges.

**Challenge 1: programming effort required to support generic side tasks.** Typically, GPU workloads are implemented based on the assumption that they have access to the full GPU and can run continuously until they finish execution. However, bubbles are *intermittent* and largely vary in duration, as in Section 2.1. A side task should be tailored to bubble patterns — the side task automatically pauses or resumes when a bubble ends or starts. Customizing the training framework to embed side tasks is conceptually feasible but limits the flexibility of implementing and executing generic GPU workloads, much like the limitations from prior work on co-running specialized procedures (Osawa et al., 2023; Thorpe et al., 2023).

**Challenge 2: limiting the impact of side tasks.** LLM training can span months on expensive high-end GPUs and cost millions of dollars (Zhang et al., 2022). Even with side tasks placed in the under-utilized bubbles, they may still interfere with pipeline training, significantly increasing the cost of LLM training and offsetting the benefit of running side tasks. However, limiting the impact of side tasks is not trivial. As the shape of bubbles varies, naively implementing side tasks may consume more resources than bubbles have — exceeding the duration of bubbles or even crashing the main task due to excessive GPU memory allocation. Ideally, bubbles should be utilized without impacting the more expensive and prioritized LLM training task.

## 3 DESIGN OF FREERIDE

FreeRide is our system that addresses the aforementioned challenges in utilizing bubbles in pipeline training to serve generic GPU side tasks. It includes two programming interfaces, an automated profiler, and FreeRide runtime consisting of a side task manager and multiple side task workers. The programming interfaces reduce the engineering effort to implement side tasks that fit into bubbles, and the automated profiler obtains the GPU resource consumption of side tasks, which is used by FreeRide runtime to minimize the overhead of side tasks on pipeline training.

Figure 2 depicts the workflow of FreeRide. First, programmers adapt their side task implementation using FreeRide's programming interfaces (step ❶). FreeRide then automatically generates a profile of the side task's characteristics (step ❷), which is submitted with the side task to the side task manager of FreeRide (step ❸). During pipeline training, the side task manager continuously adds bubbles from the instrumented training framework to FreeRide (step ❹). The side task manager assigns side tasks to workers that are deployed in Docker containers (Bernstein, 2014) based on memory allocation of pipeline training and the characteristics of side tasks (step ❺), and starts/pauses side tasks based on the available bubbles (step ❻). The side tasks access GPUs through MPS (Nvidia, 2024d) (step ❼).

In the remainder of this section, we introduce how FreeRide addresses the challenge to implement side tasks in Section 3.1, and how FreeRide minimizes the impact on pipeline training in Section 3.2.

## 3.1 PROGRAMMING OF SIDE TASKS

To address the challenge in programming effort required to support generic side tasks, we first make an important observation, that GPU workloads are not monolithic, and that they can be often divided into smaller, repeated *steps* with largely predictable per-step duration, such as epochs in model training, iterations in graph analytical workloads (Page et al., 1998), and steps to process each image in image-processing workloads (Nvidia, 2019). On the other hand, bubbles also demonstrate repeating and predictable patterns, as discussed in Section 2.1.

With these observations in mind, we abstract the life cycle of side tasks using a state machine model. The execution of side tasks within bubbles can be implemented as state transition functions. We then design programming interfaces based on this abstraction. They are discussed below.

As shown in Figure 3(a), we abstract a side task using a state machine model with five states and six state transitions. The five states capture the life cycle of a side task, from process creation to process termination, and correspond to different uses of hardware resources, e.g., GPU memory and GPU execution time. The six state transitions are used by the programmer to implement the user-defined logic of a side task, e.g., allocating or releasing hardware resources or performing computation on GPU. Once the side task is implemented, FreeRide automatically handles the state transitions at runtime. Next, using model training as an example, we discuss the states and state transitions.

• `SUBMITTED`. This state means that FreeRide has profiled a task and submitted it to the side task manager, but the side task worker has not created the side task process yet. State transition `CreateSideTask()` happens automatically after the side task manager assigns a side task to a worker and the worker creates the side task process. For a model training side task, this is where the process is first created.

• `CREATED`. In this state, the worker has created the side task process, and this process has loaded its context to the main memory but not to the GPU memory. For model training, in this state, the side task process has already created and initialized variables in CPU memory, e.g., the dataset, the data loader, and the loss function. However, the side task process will not load them into GPU memory until the side task manager initiates the state transition `InitSideTask()` which indicates the completion of side task initialization.

• `PAUSED`. This state is where the side task starts to use GPU memory. For model training, the side task process has loaded its context, e.g., model weights and optimizer states, in the GPU memory. However, this process waits in the `PAUSED` state until the side task manager transitions its state to `RUNNING` through `StartSideTask()`.

• `RUNNING`. In this state, the side task executes the step-wise GPU workload. Referring to the example above of the model training side task, this step involves reading the next batch, computing the output and loss, updating the model weights, and resetting the optimizer states. The side task iteratively enters the `RunNextStep()` state transition to execute these steps until the side task manager transitions its state through `PauseSideTask()`. Therefore, in this state, the side task process uses both the GPU memory and the GPU SMs.

• `STOPPED`. This state marks the end of the life cycle of a side task, where the side task process releases all of its hardware resources and terminates. It can be transited from states `CREATED`, `PAUSED`, and `RUNNING` through `StopSideTask()` initiated by the side task manager.

Figure 3(b) shows state transitions of a side task in Stage 0 of Figure 1. Initially, the side task is in the `PAUSED` (P) state. After four FP operations in the main training workload have finished, a bubble starts and the side task manager initiates `StartSideTask()` to transit the side task to the `RUNNING` (R) state. After the first bubble ends, the side task manager pauses the side task via `PauseSideTask()`. Then, the main training workload has BP operations and bubbles interleaved, leading to back-and-forth transitions between `PAUSED` and `RUNNING` states of the side task.

Given the state machine abstraction, the next step is to implement side tasks, which have two requirements. First, the programmer should be able to implement the side task in a way that can pause at the end of a bubble and resume at the start of the next bubble. Second, the side task should

be able to communicate with the side task manager to receive state transition RPCs for pausing and resuming. To lift programming burdens, FreeRide provides two programming interfaces, the *iterative interface*, and the *imperative interface*. The iterative interface is the preferred one for side tasks in FreeRide. It requires the side task to be step-wise, e.g., model training, and provides the lowest performance overhead. For other side tasks that cannot be explicitly implemented step-wise, the imperative interface is the fallback solution. It offers better versatility to support (almost) generic GPU workloads at the cost of higher performance overhead. Both interfaces incorporate the communication of side tasks with other components, and the programmer only has to apply a few lines of changes. We leave the details of programming interfaces as well as examples in Appendix A.1.

## 3.2 Minimizing the Impact on Pipeline Parallel Training

To address the challenge of limiting the impact of side tasks on the main pipeline training workload taking three approaches, FreeRide first leverages offline profiling to understand the shapes of bubbles and characteristics of newly submitted side tasks (Section 3.2.1). Based on the profiling results, FreeRide employs one *side task manager* and multiple *side task workers*, one for each GPU. The side task manager assigns the newly submitted side task to one of the side task workers with enough GPU memory, and initiates state transitions of side tasks through remote procedure calls (RPCs) at the start and end of each bubble, which are reported by DeepSpeed that we instrument (Section 3.2.2). FreeRide further employs CUDA MPS (Nvidia, 2024d) and a twofold mechanism to prevent side tasks from excessively allocating GPU memory or not pausing correctly (Section 3.2.3).

### 3.2.1 Profiling Bubbles and Side Tasks

**Bubbles.** To know the shapes of bubbles, FreeRide runs DeepSpeed, monitors its estimated SM occupancy and GPU memory consumption through the PyTorch profiler (PyTorch, b), and automatically measures each bubble's duration and available GPU memory. Since the pipeline schedule determines bubbles, this offline profiling is done only once for each model and pipeline scheduling on the same hardware platform.

**Side tasks.** After the programmer implements the side task, FreeRide profiles it with the automated profiling tool for its performance characteristics of GPU memory consumption and per-step duration, which FreeRide uses for side task management and GPU resource limit. For side tasks implemented using the iterative interface, this procedure is fully automated. The profiling tool runs the side task, monitors its GPU memory consumption, and records the timestamps at the start and end of `RunNextStep()` for the per-step duration. For side tasks implemented using the imperative interface, the tool profiles GPU memory consumption in the same way. However, since the side task is not implemented step-wise, the automated profiling tool does not measure the per-step duration.

### 3.2.2 Side Task Management

FreeRide's side task management has two main roles. First, upon receiving a new side task, the side task manager assigns it to a suitable side task worker. Second, when the pipeline training system adds bubbles to the side task manager, the side task manager initiates the state transitions of side tasks (Figure 3(a)) through RPCs. This way, the side tasks are only served during bubbles and do not compete for GPU resources with the main pipeline training workload.

To do so, when the side task manager receives a new side task, it first selects all workers whose bubbles have enough available GPU memory for the side task. Then, it assigns the side task to worker with the least side tasks waiting to be served. During pipeline training, the side task manager periodically checks whether any bubble has just started or expired. If a bubble has just started, the side task manager starts the execution of the corresponding side task, or adds a new side task to the bubble's worker. If a bubble has just expired, the side task manager pauses the corresponding side task. We demonstrate the detailed side task management algorithms in Appendix A.2.

### 3.2.3 GPU Resource Limit

**GPU Memory.** FreeRide leverages MPS to impose GPU memory limit on side tasks. I.e., when a worker creates a side task, it sets GPU memory limits using MPS. The side task process triggers an out-of-memory (OOM) error when its memory consumption exceeds the limit, but other processes

remain unaffected. However, FreeRide is also compatible with other mechanisms for limiting GPU memory, e.g., multi-instance GPU (MIG) (Nvidia, 2024c) or manually implemented accounting through intercepting CUDA kernel calls (Strati et al., 2024).

**GPU Execution Time.** FreeRide limits GPU execution time using two mechanisms. (1) *The program-directed mechanism* is tailored for the iterative interface. When the side task manager makes an RPC to initiate `StartSideTask()` state transition of a side task, it also sends the end time of this bubble to the side task. After the state transition finishes, the side task enters the `RUNNING` state. Before the side task automatically starts `RunNextStep()`, the program-directed mechanism checks if the remaining time of the bubble is enough for the side task to execute the next step. The side task will only execute the next step if the remaining time exceeds the per-step duration. (2) *The framework-enforced mechanism* supports side tasks implemented using the imperative interface and is also a fallback mechanism for the iterative interface. After the side task manager initiates the `PauseSideTask()` state transition for a side task, it waits for a short grace period before checking the *last paused timestamp* — a timestamp maintained by the interface that records the last time the side task was paused. If this timestamp is not updated after the state transition begins, the side task manager assumes that the interface failed to pause the side task correctly and subsequently instructs the corresponding worker to terminate the side task process using `SIGKILL`. The side task initialization, `InitSideTask`, which runs only once throughout the life cycle of a side task, is also protected by this mechanism.

# 4 EVALUATION

In this section, we evaluate the benefits and overhead of using FreeRide to serve side tasks.

## 4.1 METHODOLOGY

We describe the experimental setup of our evaluation.

**Server setup.** We use a main server (Server-I) with four RTX 6000 Ada GPUs each with 48 GB of GPU memory to evaluate all pipeline training workloads and side tasks. We use a second server (Server-II) with an RTX 3080 GPU with 10 GB of memory to run side tasks separately. Due to the global shortage of cloud GPUs, we quote prices from a community cloud vendor RunPod (2024) that has GPUs available. The prices of the two servers are $P_{\text{Server}-\text{I}} = \$3.96/\text{hour}$ and $P_{\text{Server}-\text{II}} = \$0.18/\text{hour}$, respectively (as of June, 2024). The price differences between higher- and lower-tier GPUs in major cloud GPU platforms are similar (Lambda, 2024; Amazon, 2024a;b). We deploy both pipeline training and side tasks in Docker 26.1.2 (Bernstein, 2014).

**Comparison points.** We evaluate FreeRide for side tasks developed with both the iterative and imperative interfaces. For comparison, we evaluate MPS (Nvidia, 2024d), where we set pipeline training with the highest priority and side tasks with a lower priority. We also evaluate a naive co-location approach by directly co-running side tasks and the main pipeline training workload on the same GPU.

**Pipeline training setup.** We train LLMs adapted from GPT2-XL (Radford et al., 2019; Karpathy, 2024; Choi et al., 2023) with model sizes 1.2B, 3.6B, and 6B with DeepSpeed 0.12.2 (DeepSpeed, 2023) in a 4-stage pipeline on Server-II (stages 0—3 in Figure 1). We always maximize the micro-batch size (until just before OOM) to make full use of GPU memory during training.

**Side task workloads.** We implement 3 types of side tasks: model training, graph analytics, and image processing using both the iterative and the imperative interfaces of FreeRide. *Model training* side tasks include ResNet18, ResNet50, and VGG19. We use the out-of-the-box models from PyTorch (PyTorch, a) and implement the training procedure ourselves. *Graph analytics* side tasks are adapted from Gardenia (Xu et al., 2019). It includes PageRank (PR) based on the PageRank algorithm (Page et al., 1998) and Graph SGD (SGD) which uses stochastic gradient descent to solve matrix factorization (Koren et al., 2009), both using the Orkut dataset (Yang & Leskovec, 2012). The *image processing* (Image) side task resizes an input image and adds a watermark, which we adapt from Nvidia's example (Nvidia, 2019).

Table 1: Time increase $I$ (lower the better) and cost savings $S$ (positive=benefit, negative=loss, higher the better) of running DeepSpeed with side tasks using RTX 3080 as the proxy.

| Side task | FreeRide Iterative | | FreeRide Imperative | | MPS | | Naive | |
|---|---|---|---|---|---|---|---|---|
| | $I$ % | $S$ % | $I$ % | $S$ % | $I$ % | $S$ % | $I$ % | $S$ % |
| ResNet18 | 0.9 | 6.4 | 2.2 | 6.0 | 16.8 | -1.5 | 49.8 | -30.7 |
| ResNet50 | 0.9 | 5.3 | 3.8 | 3.9 | 19.8 | -5.1 | 61.9 | -44.0 |
| VGG19 | 0.9 | 3.9 | 5.0 | 1.4 | 21.4 | -9.1 | 53.4 | -39.7 |
| PageRank | 1.0 | 11.1 | 2.5 | 16.4 | 17.3 | 3.5 | 45.1 | -16.0 |
| Graph SGD | 1.2 | 11.8 | 4.1 | 22.8 | 231.0 | -26.7 | 62.4 | -9.1 |
| Image | 1.4 | 5.7 | 2.7 | 6.1 | 9.5 | 7.2 | 46.0 | -29.3 |
| Mixed | 1.1 | 10.1 | 4.3 | 11.0 | 24.8 | 0.2 | 64.3 | -35.5 |

**Metrics.** We use the *time increase $I$* and *cost savings $S$* in Dollars due to side tasks as metrics. Time increase describes the performance overhead of co-locating side tasks with the main pipeline training workload. It is the ratio of extra time of pipeline training with side tasks, compared with the original DeepSpeed without any side tasks, defined as

$$I = \frac{T_{\text{withSideTasks}} - T_{\text{noSideTask}}}{T_{\text{noSideTask}}} \quad .$$

Cost savings describe the benefits of running side tasks. Since we cannot directly compare the throughput of different side tasks and the main pipeline training workload, we use their cost (dollars spent on GPUs) as a proxy. First, we define the cost of pipeline training without side tasks as $C_{\text{noSideTask}}$, and that with side tasks as $C_{\text{withSideTasks}}$. Then, we define the cost of running the same side tasks on dedicated GPUs as $C_{\text{sideTasks}}$. Finally, we define cost savings $S$ as

$$S = \frac{C_{\text{sideTasks}} - (C_{\text{withSideTasks}} - C_{\text{noSideTask}})}{C_{\text{noSideTask}}} \quad .$$

We demonstrate the detailed definitions in Appendix A.3.

## 4.2 PERFORMANCE EVALUATION

We run DeepSpeed to train a 3.6B model for 128 epochs with side tasks from Section 4.1 and compare the performance overhead, i.e., time increase ($I$) and cost savings ($S$) of using FreeRide with the two interfaces and the two comparative methods (as in Section 4.1) using RTX 3080 as the proxy. For model training side tasks, we set the batch size to 64. We run the *same side task* in all workers if they have enough GPU memory. We also run a *mixed workload* with 4 side tasks: PageRank, ResNet18, Image, and VGG19, each in one worker corresponding to the GPU of stages 0—3 in Section 4.1, respectively.

The results are summarized in Table 1. FreeRide consistently exhibits lower overhead than the comparative methods, showing only a 1.1% average time increase while achieving 7.8% average cost savings through side tasks using the iterative interface. The imperative interface achieves comparable cost savings but with a higher overhead as it relies on the less efficient framework-enforced mechanism to limit the side task's execution time (Section 3.2.3). In comparison, the average time increase and cost savings for MPS are 48.7% and -4.5%, and for Naive are 54.7% and -29.2%. Their *negative* cost savings indicate that these approaches can increase the total cost. Notably, the time increase of Graph SGD with MPS is as high as 231.0%. This anomaly is due to Graph SGD's high compute intensity. We conclude that FreeRide effectively utilizes bubbles in pipeline training for serving side tasks. While the comparative methods can utilize bubbles, unlike FreeRide, they are not designed for this purpose. Thus, they are inefficient in using bubbles, leading to higher costs.

## 4.3 BUBBLE TIME BREAKDOWN

In Figure 4, we present a breakdown of bubble utilization in FreeRide under the iterative interface. *No side task: OOM* means that some bubbles are unused due to their limited available GPU memory. E.g., the GPU memory consumption of VGG19 or the Image side task is larger than the GPU memory of bubbles in stages 0 and 1, so they cannot use half of the bubble time. *No side task: insufficient time* refers to idle time because the remaining time of a bubble is not enough for the next step of the

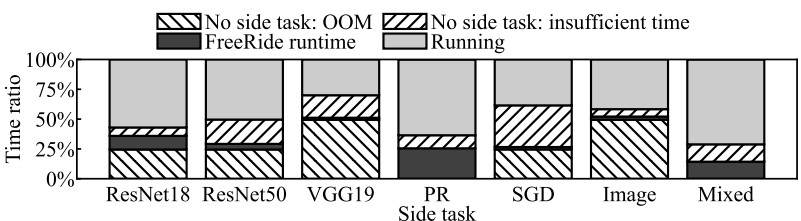

Figure 4: Bubble time breakdown.

side task. *FreeRide runtime* is the time consumed by running FreeRide, including the interface code and the side task manager. Most of the bubble time with enough available GPU memory size is used by side tasks. For side tasks with shorter per-step durations, e.g., PageRank, the ratio of FreeRide runtime is higher because more iterations of the iterative interface are executed. In contrast, side tasks with longer per-step durations have lower bubble utilization because of insufficient time.

We also conducted sensitivity in Appendix A.4 to demonstrate the superiority of FreeRide in different pipeline training settings, and studied the effectiveness of GPU resource limit mechanisms that keep of FreeRide that keep the time increase low in Appendix A.5.

## 5 DISCUSSION AND RELATED WORK

**Security.** Prior GPU sharing solutions tend to prioritize efficiency and assume a safe environment. E.g., Orion assumes that co-located GPU workloads are in the same trust domain (Strati et al., 2024). FreeRide provides the same security and isolation guarantees as the lower-level system it is built upon. It incorporates MPS to limit GPU memory which provides separate GPU address spaces (Nvidia, 2024b) for pipeline training and side tasks, and Docker for environment isolation (Docker, 2024). Orthogonally, security for co-located GPU workloads is an active research area (Liu et al., 2019; Zhang et al., 2024). We expect future work to co-design security with efficient GPU sharing.

**Side task management.** By implementing different strategies in its side task manager, FreeRide can incorporate more sophisticated management, e.g., co-locating multiple side tasks with various performance characteristics in the same worker to improve the utilization of bubbles (Liu et al., 2022b) or serving side tasks with fairness or performance guarantees (Ghodsi et al., 2011).

**Scalability.** FreeRide can be extended for better scalability. As FreeRide implements communications among its components using RPCs, it can be easily extended to distributed settings with side tasks on multiple servers. FreeRide can also be extended for multi-GPU side tasks, e.g., distributed training and big data processing (Liu et al., 2022a), by launching workers with access to multiple GPUs.

**Other ML accelerators.** This work targets GPUs due to their widespread accessibility. FreeRide's mitigation for bubbles fundamentally applies to other ML accelerators (Jouppi et al., 2017; Meta, 2023), provided that the platform has isolation and resource limit options for each process. We anticipate future work to incorporate the approach of FreeRide with other ML platforms.

## 6 CONCLUSION

We propose FreeRide, a system to harvest the bubbles in pipeline parallelism to serve generic GPU side tasks. It provides programming interfaces that abstract the life cycle of a side task as different states of a state machine and allows programmers to implement side tasks with little engineering effort. The side task manager and side task workers manage bubbles and side tasks and reduce the performance overhead of side tasks on pipeline training. Our evaluation shows that, on average, FreeRide achieves 8% cost savings for long-running and expensive pipeline training with a negligible performance overhead of only about 1%.

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

# A APPENDIX

## A.1 USE OF SIDE TASKS INTERFACE

This section describes FreeRide's iterative and imperative interface mentioned in Section 3.1 in detail.

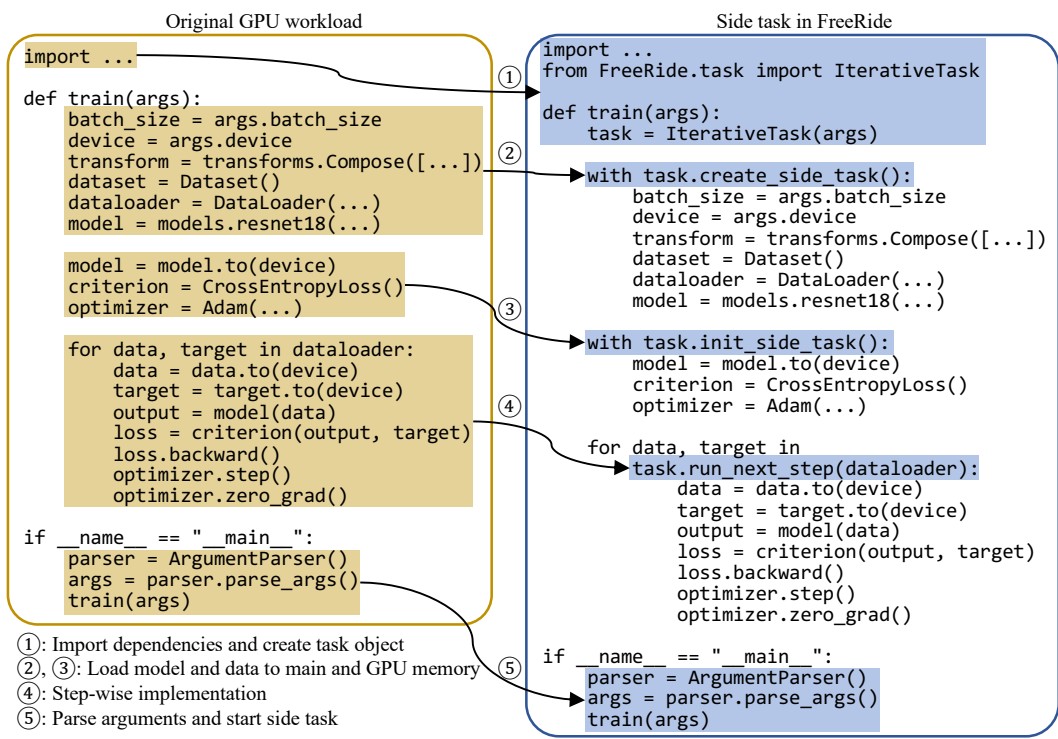

Figure 5: Example of implementing ResNet18 training using the iterative interface of FreeRide.

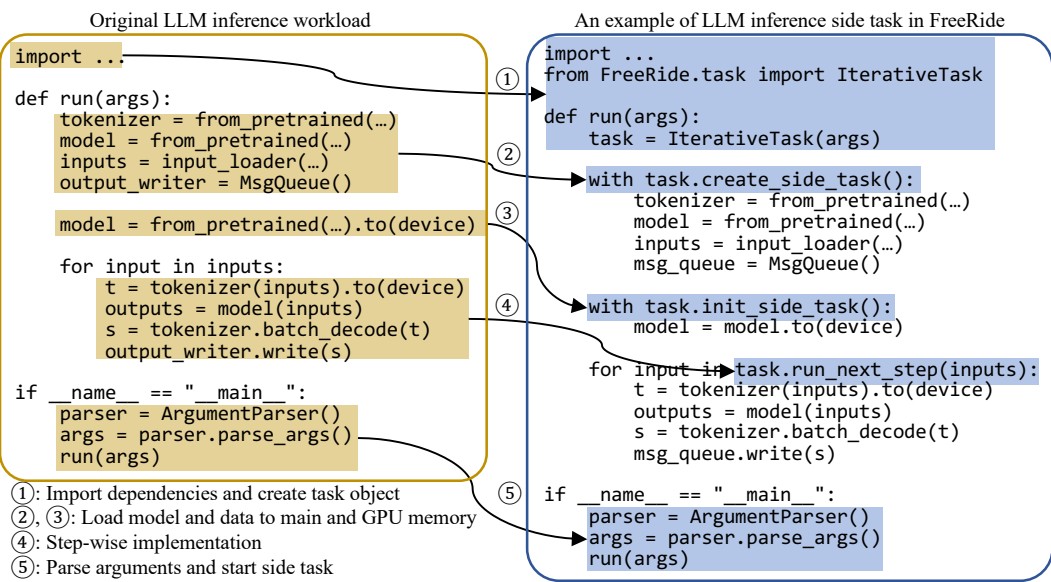

Figure 6: Example of LLM inference using the iterative interface of FreeRide.

**Iterative programming interface.** In Figure 5 and Figure 6, we present examples of implementing side tasks to train ResNet18 and to do LLM inference using the iterative interface of FreeRide in

Python. As demonstrated by these two examples, the flexibility and versatility of FreeRide enables the user to implement various side tasks with little engineering effort. Less important lines such as importing dependencies and parsing arguments are simplified. Porting this example involves mainly five steps. Step ①: import FreeRide dependencies and inherit the iterative interface class, which includes an implementation for the state machine abstraction, communication with the side task manager, and the program-directed mechanism to limit the GPU execution time. The programmer only has to migrate the implementation of the original GPU workload to the interface. Steps ② and ③: implement the side task initialization in 2 state transition functions, `CreateSideTask()` and `InitSideTask()`, to load the context into main memory and GPU memory respectively. Step ④: wrap the original loop implementation with `RunNextStep()`. Step ⑤: the main function handles argument parsing and runs the side task interface.

Most of the modifications are trivial, e.g., wrapping implementations with side task state transition functions in Step ②, ③, and ④, which are required by Python. Aside from this, the programmer can directly copy the important logic, e.g., loading the dataset and training the model, from the original implementation. In addition, if the programmer customizes the model architecture instead of using the publicly available ones, the model implementation also does not require modification.

**Imperative programming interface.** This interface does not require the programmer to implement the side task in a step-wise way. Therefore, instead of implementing the side task in multiple functions (steps ② — ④), the programmer can merge them in `RunGpuWorkload()`. However, this approach trades performance for less programming effort, as pausing side tasks through the framework-enforced mechanism incurs more overheads. When the side task manager initiates `PauseSideTask()` state transition via an RPC at the end of a bubble, even though the CPU process of the side task is paused by the framework-enforced mechanism (Section 3.2.3) after the state transition, CUDA kernels that have already started cannot be paused because they are asynchronous Nvidia (2024a). As a result, these CUDA kernels will overlap with pipeline training, causing a higher performance overhead than the iterative interface.

## A.2  Side Task Management Algorithms

In this section, we present the side task management algorithms mentioned in Section 3.2.2. To keep track of side tasks and workers, the side task manager maintains the following fields for each worker, used by Algorithms 1 and 2 for side task management:

- *GPUMem*: the available GPU memory size.

- *TaskQueue*: the queue of side tasks ordered by submission timestamps.

- *CurrentTask*: the side task that is currently served.

- *CurrentBubble*: the bubble that is currently valid.

---

**Algorithm 1:** Procedure upon a new side task.

---
1: **Input:** new side task $Task$, workers' metadata $Workers$
2: $MinNumTasks \leftarrow \infty, SelectedWorker \leftarrow None$
3: **for all** $Worker$ **in** $Workers$ **do**
4:   **if** $Worker.GPUMem > Task.GPUMem$ **then**
5:     $NumTasks \leftarrow Worker.GetTaskNum()$
6:     **if** $NumTasks < MinNumTasks$ **then**
7:       $MinNumTasks \leftarrow NumTasks, SelectedWorker = Worker$
8: **if** $SelectedWorker \neq None$ **then**
9:   $SelectedWorker.Add(Task)$
10: **else**
11:   $RejectSideTask()$

---

Algorithm 1 describes how the side task manager assigns side tasks to workers. When the side task manager receives a new side task together with its GPU memory requirement (through profiling, Section 3.2.1), it first filters out all workers with enough available GPU memory (lines 2—3). Then, from these workers, it selects the one with the smallest number of tasks (lines 4—7). If the side task

---

**Algorithm 2:** Managing bubbles and side tasks.

1: **Input:** workers' metadata $Workers$
2: **while** $SideTaskManagerIsRunning$ **do**
3:    **for all** $Worker$ **in** $Workers$ **do**
4:      **if** $Worker.CurrentBubble \neq None$ **and** $Worker.CurrentBubble.HasEnded()$ **then**
5:        **if** $Worker.CurrentTask \neq None$ **then**
6:          $Worker.CurrentTask.PauseSideTask()$
7:        $Worker.CurrentBubble \leftarrow None$
8:      **if** $Worker.HasNewBubble()$ **then**
9:        $Worker.UpdateCurrentBubble()$
10:       **if** $Worker.CurrentTask = None$ **then**
11:         **if** $Worker.TaskQueue.IsEmpty()$ **then**
12:           *continue*
13:         $Worker.CurrentTask \leftarrow Worker.TaskQueue.Next()$
14:       **if** $Worker.CurrentTask.IsCreated()$ **then**
15:         $Worker.CurrentTask.InitSideTask()$
16:       **else if** $Worker.CurrentTask.IsPaused()$ **then**
17:         $Worker.CurrentTask.StartSideTask()$

---

manager has selected a worker, it will assign the side task to that worker (lines 8—9). Otherwise, it will reject the side task because of insufficient GPU memory (line 11).

Algorithm 2 describes how the side task manager manages bubbles and side tasks during pipeline training. The side task manager iterates through all workers (line 2). If *CurrentBubble* has just ended for a worker, the side task manager will pause *CurrentTask* of the worker and clear *CurrentBubble* (lines 3—7). Upon a new bubble, the side task manager updates the *CurrentBubble* of this worker (lines 8—9). It then checks if the worker has a *CurrentTask*. If not, it will select the one with the smallest submission timestamp from *TaskQueue* as *CurrentTask* (lines 10–13). After that, the side task manager initiates `InitSideTask()` if the newly added *CurrentTask* is in `CREATED` state (lines 14—15); otherwise, its state is `PAUSED` and the side task manager initiates `StartSideTask()` (lines 16—17).

### A.3 DETAILED DEFINITION OF METRICS

In this section, we describe the detailed definition of *time increase $I$* and *cost savings $S$*. Time increase describes the performance overhead of co-locating side tasks with the main pipeline training workload. It is defined as

$$I = \frac{T_{\text{withSideTasks}} - T_{\text{noSideTask}}}{T_{\text{noSideTask}}} \quad .$$

For cost savings, we define the cost of pipeline training without side tasks as

$$C_{\text{noSideTask}} = P_{\text{Server-I}} \times T_{\text{noSideTask}} \quad ,$$

the cost of pipeline training with side tasks as

$$C_{\text{withSideTasks}} = P_{\text{Server-I}} \times T_{\text{withSideTasks}} \quad ,$$

and the cost of running the same side tasks on dedicated GPUs as

$$C_{\text{sideTasks}} = \sum_{\text{Each sideTask}} P_{\text{Server-II}} \times \frac{W_{\text{sideTask,Server-I}}}{Th_{\text{sideTask,Server-II}}} \quad .$$

where $W_{\text{sideTask,Server-I}}$ is the work done by a side task on Server-I, e.g., the number of epochs for model training side tasks, the number of iterations for graph analytics side tasks, and the number of images for the image processing side task. $Th_{\text{sideTask,Server-II}}$ is the throughput of running the same side task on Server-II, which we measure by running side tasks individually on Server-II. Finally, we define the cost savings $S$ as

$$S = \frac{C_{\text{sideTasks}} - (C_{\text{withSideTasks}} - C_{\text{noSideTask}})}{C_{\text{noSideTask}}} \quad .$$

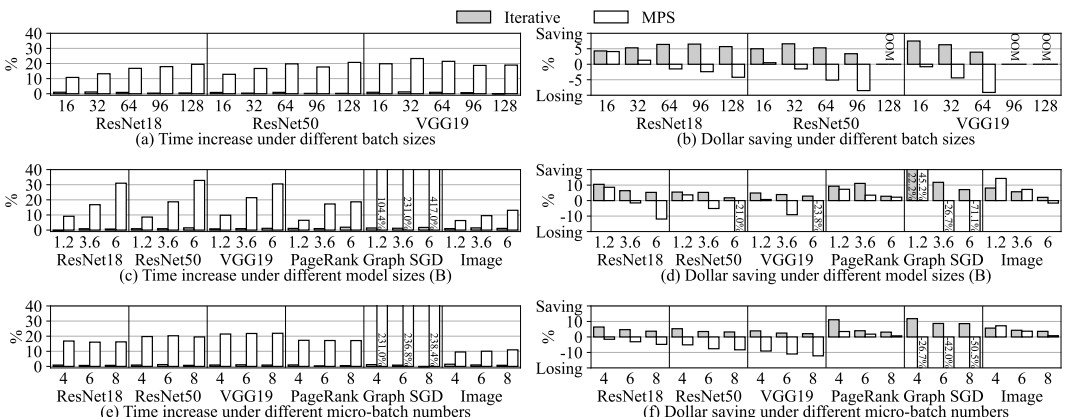

Figure 7: Sensitivity studies of FreeRide.

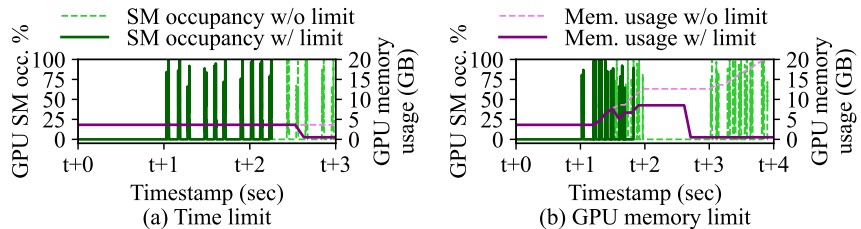

Figure 8: Demonstration of GPU resource limit in FreeRide.

### A.4 SENSITIVITY STUDY

This section describes the sensitivity study which demonstrates that FreeRide can achieve superior time increase and cost savings compared with MPS in different settings. We change the side task batch size, DeepSpeed model size, and DeepSpeed micro-batch numbers of different side tasks, and study the time increase and cost savings of FreeRide with the iterative interface.

**(1) Varying batch sizes.** Figure 7(a) and (b) include model training side tasks under variable batch sizes. Other side tasks are not included as they do not run with batch sizes. *OOM* means that the GPU in Server-II does not have enough GPU memory for the configuration, so the cost savings cannot be calculated. FreeRide has low performance overheads, with around 1% increase in execution time, and cost savings of 3.4% – 7.5%.

**(2) Varying model sizes.** In Figure 7(c) and (d), the performance overheads of FreeRide range from -0.7% to 1.9%, and cost savings range from 1.8% to 22.2%. The main reason is the shorter bubble durations when training larger models as the main workload, which was also shown in Figure 1.

**(3) Varying micro-batch numbers.** In Figure 7(e) and (f), the performance overhead of FreeRide increases from -0.4% to 1.5%, and cost savings reduces from 2.1% to 11.8%. When the micro-batch number increases, because of the lower bubble rate (Section 2.1), the cost savings decrease.

### A.5 EFFECTIVENESS OF GPU RESOURCE LIMIT

This section demonstrates the effectiveness of GPU resource limit mechanisms of FreeRide. We use training ResNet18 as an example.

**Side task execution time limit.** Figure 8(a) demonstrates a case where the side task does not pause after the bubble that ends at $t + 2$. With GPU resource limit, as shown by the green and purple curves, the worker terminates the side task after a grace period via the framework-enforced mechanism.

**Side task GPU memory limit.** Figure 8(b) illustrates another case where the side task keeps allocating GPU memory despite its 8 GB limit. Without FreeRide's GPU resource limit mechanism,

the side task's GPU memory allocation is only capped by the physical memory limit of the GPU, potentially interfering with the main training workload. With GPU resource limit, after the side task process exceeds its 8 GB GPU memory limit, it is terminated to release GPU memory.

