# OpenReview forum: "FreeRide: Harvesting Bubbles in Pipeline Parallelism"
_ICLR.cc/2025/Conference — ICLR 2025 Conference Withdrawn Submission_

### Official Review · Reviewer_pb5V · 2024-11-03

**Soundness:** 2
**Presentation:** 2
**Contribution:** 2
**Rating:** 5
**Confidence:** 3

**Summary:**

GPUs are sometimes idle during LLM training. This paper proposed a framwork to schedule side tasks during the idle time (bubbles) to improve utilitzation. It requires careful scheduling of side tasks to avoid using too much memory or too much compute to degrade the the LLM training job itself.

From UX perspective, the paper proposed 2 approach. The one with more details are iterative programming interface. Users can annotate their training loop with context managers to create/init/step side tasks.

From performance perspective, it's critical to understans when the bubble will happen and how long it lasts. It also critical to undersand how much memory and runtime the side task need per step. It uses pytorch profiler traces for such purpose. It also enforce maximum of gpu execution time and memory usage for side tasks.

The author breifly talked about security, fairnesss among side tasks, scalability of the framework itself.

**Strengths:**

* Significance: scheduling side tasks during bubbles is a generic way to improve gpu utilization. Bubbles exist not only in pipeline parallelsim but in any training jobs. One way to deal with bubble is to optimize the training itself. This paper focused on an alternative to schedule 3rd parity tasks during bubbles
* Clear programming interface: user can annotate their trainin loop with context managers at init and step, according to the the iteractive programming interface. The amount of code change is not too complicated

**Weaknesses:**

* Less details in estimating bubble shapes: It briefly mentioned the usage of pytorch profiler traces. But llm training job itself can be quite dynamic. bubble shapes can change due to stragglers, variable sequence length, and multi-tenant usage (including shared usage of CPUs). It would be great if the author could explain more on how indicative the profilter traces are to predict bubbles shape for future runs
* Less details in fault tolerance: practically speaking, side tasks can be any 3rd party tasks and are error prone. It remains unclear how fatal errors from side tasks are kept self contained without affecting the main LLM training job itself. Example fatal erros are illegal memory access and NCCL timeouts.

**Questions:**

According to the iterative programming interface, side tasks and main LLM training jobs seems to be sharing memory and cuda streams directly. I would love to learn more details in isolating fatal error from side tasks, specifically, we can use illegal memory access and NCCL timeout as example fatal error to clarify

Side jobs can have side effects even they are executed successfully. One example is cuda caching allocator in pytorch. Side effects could icnrease the cache memory and increase memory fragmentation. Curious to learn more about how to overcome the side effects.

---

### Official Review · Reviewer_sPoA · 2024-11-03

**Soundness:** 2
**Presentation:** 3
**Contribution:** 2
**Rating:** 3
**Confidence:** 5

**Summary:**

This paper introduces FreeRide, a framework aimed at enhancing GPU utilization during idle periods ("bubbles") in LLM training with pipeline parallelism. FreeRide profiles workloads to capture computation and memory characteristics, then schedules side tasks on idle GPUs. Experimental results indicate that FreeRide can reduce overall training costs with minimal overhead.

**Strengths:**

1. FreeRide introduces an effective approach for leveraging idle pipeline periods in LLM training by assigning side tasks to available GPUs.
2. The paper is well-organized and accessible, with a clear categorization of different bubble types and reasonable methods for utilizing these idle periods.

**Weaknesses:**

1. The experimental setup is questionable. Side tasks are run on an RTX3080, a GPU with lower computational capacity, whereas a fair comparison would require using the same GPU model (e.g., RTX 6000 Ada).
2. The experiments seem to be imaginary scenarios, which may not align with current industry practices. Probably utilizing bubbles for LLM inference or serving may be a more interesting use case.
3. Pipeline parallelism is primarily used in multi-machine settings, typically in combination with 3D parallelism across machines, with pipeline parallelism spanning machines instead of the single-machine setting in the paper. Communication overhead should be addressed in such settings.
4. No discussion on model checkpoints. Frequent switching or termination of side tasks could necessitate checkpointing, which incurs additional overhead.
5. The focus is heavily on system optimization, with limited ML-specific insights, raising questions about its suitability for ICLR.

**Questions:**

1. Why were different servers used for evaluation? A fair comparison would involve the *same* server setup to (1) train both the LLM and side tasks concurrently using FreeRide, and (2) sequentially train the LLM, then dedicate all resources to the side tasks. Comparing the total training time for (1) and (2) could demonstrate FreeRide’s effectiveness in utilizing idle resources.
2. What occurs if the primary training job crashes or finishes before the side tasks? Will the side tasks automatically adjust to use all available resources to complete remaining computations?
3. How do the experiments reflect real-world scenarios? Please provide references if some companies concurrently run LLM with ResNet or graph analytics jobs.
4. Is it possible to make LLM inference/serving as a side task? How would FreeRide support this? Such a feature might increase FreeRide's practical relevance.
5. Since pipeline parallelism is typically deployed in multi-machine settings, with 3D parallelism (i.e., data, tensor, pipeline) across machines, how does FreeRide plan to handle inter-machine communication and support 3D parallelism?
6. How does FreeRide handle checkpointing for the main and side tasks? Frequent checkpointing could increase overhead and impact idle period durations.
7. Can FreeRide support models optimized with compiler techniques (e.g., torch.compile, CUDA Graphs) during training? After leveraging the interface in Figure 5, it appears challenging to integrate other optimization methods (e.g., torch.compile, ZeRO3 parallelism). How do you plan to support these additional optimizations?

---

### Official Review · Reviewer_JZkK · 2024-11-03

**Soundness:** 3
**Presentation:** 4
**Contribution:** 2
**Rating:** 5
**Confidence:** 3

**Summary:**

The paper presents FreeRide, a system designed to help users take advantage of idle periods in GPU task execution while training a model with pipeline parallelism. The authors contend that these idle periods, or "bubbles," cannot be completely eliminated; however, they can be utilized to make GPUs accessible for other computations during times when they are not in use.

**Strengths:**

- Original piece of work
- Experiment methodology is sound
- All components are explained well

**Weaknesses:**

The total end-to-end cost savings from running additional tasks during pipeline bubbles are relatively modest, estimated to be under 10%. In practice, this figure could be even lower unless a workload consists of small tasks with precise memory requirements that align perfectly with the available bubbles. Such gains may not be sufficient to justify the complexity of implementing FreeRide. Additionally, if the implementation is not perfect, it could result in overheads that negate any cost savings achieved.

Given this context, it would be helpful if the authors could clarify the target users or specific products/applications for which: (1) there is a need to train a large model using pipeline parallelism while simultaneously running a smaller task that can fit within the large model's bubbles, and (2) an 8% cost savings would be significant enough to be worth considering.

**Questions:**

The introduction states that up to 40% of the total training time of a large language model (LLM) may be lost due to inefficiencies known as "bubbles." However, the authors also reference studies by Fan et al. (2021), Liu et al. (2023), and Qi et al. (2024), which propose techniques to minimize bubble time. After implementing these techniques, what is the average percentage of total training time that continues to be wasted on bubbles?

---

### Note · Authors · 2024-12-11

I have read and agree with the venue's withdrawal policy on behalf of myself and my co-authors.